# Public perceptions of physician-pharmaceutical industry relationships and trust in physicians

**Sayaka Saito[1], Kei Mukohara[2]\*, Kazuhiro Shimomura[3,4], Kenta Murotani[5]**

**1** Department of General Medicine, National Hospital Organization Kasumigaura Medical Center, Tsuchiura, Ibaraki, Japan, **2** Department of General and Family Medicine, Kurume University Medical Center, Kurume, Fukuoka, Japan, **3** Department of Biostatistics, Graduate School of Medicine, Kurume University, Kurume, Fukuoka, Japan, **4** Department of Pharmacy, Aichi Cancer Center Hospital, Nagoya, Aichi, Japan, **5** Biostatistics Center, Kurume University, Kurume, Fukuoka, Japan

\* mukohara_kei@kurume-u.ac.jp

**Data Availability Statement:** All relevant data are within the paper and its Supporting Information files.

**Funding:** This work was supported by Japan Society for the Promotion of Science (JSPS)

## Abstract

### Background

In Japan, as elsewhere, physicians meet with and receive gifts from pharmaceutical representatives (PRs). This study aimed to clarify the Japanese public perceptions of physicians' relationships with PRs, examine the association between these perceptions and their trust in physicians, and compare the public's and physicians' awareness, acceptance, and perceptions of the influence of physician-PR relationships.

### Methods

A cross-sectional, self-administered, anonymous, internet-panel survey was conducted involving 1,000 participants from the general public. The survey implementation was contracted to Cross Marketing Inc.

### Results

The mean age of the 1000 participants was 44.8 years (standard deviation 18.3). Forty-eight percent were female. Many of our participants were unaware of certain physician-PR relationships. The public was more acceptable with physicians' receiving stationery and/or medical textbooks and attending promotional drug seminars at their workplaces compared with receiving meals at restaurants. Many thought that physicians' involvement in promotional activities influenced their prescribing habits and estimated that the majority of physicians received office stationery and meals from PRs. They were divided as to whether they would like to know about their physicians' relationships with the industry. Factors associated with higher trust in physicians included participants being 65 years or older, having a primary care physician, being in better health, the belief that physicians' involvement in promotional activities is acceptable, and their high estimate that physicians are not receiving gifts from PRs. Compared to the physicians, the public had lower awareness of and was more accepting of physicians' involvement in promotional activities. Meanwhile, the public believed that

KAKENHI Grant Numbers JP 20H03920 and JP 22H04923 (CoBiA). The funders had no role in study design, data collection and analysis, decision to publish, or preparation of the manuscript.

**Competing interests:** The authors have declared that no competing interests exist.

physician-PR relationships influenced physicians' prescribing habits more than the physicians themselves.

## Conclusion

Our survey provided insights into Japanese public perceptions of physician-pharmaceutical industry relationships and their impact on trust in physicians. Physicians should be aware of these perceptions and carefully consider how to foster appropriate relationships with the industry.

## Introduction

In Japan, as elsewhere, physicians are known to meet with and personally receive gifts from pharmaceutical representatives (PRs) [1]. In theory, gifts from the pharmaceutical industry put physicians at risk of losing the trust of patients and society in general [2]. The American College of Physicians suggests that it is useful for physicians to ask themselves whether they are willing to have these relationships generally known [3]. Physicians need to know how patients and the general public perceive the relationships between physicians and the pharmaceutical industry. In our previous qualitative study, we found that physicians would change the way they interact with PRs if they were aware of how their patients perceive such relationships [4].

Several surveys have shown how patients and the public perceive the relationships between physicians and PRs [5–11]. The proportions of patients and the public who are aware of specific physician-PR relationships have varied significantly across these studies. The results have also diverged regarding the appropriateness and acceptability of gifts. Notably, in comparison with physicians' own opinions, patients in a United States (U.S.) study perceived that gifts had a greater impact on physicians' prescribing behaviors [11]. In a Japanese study, 64.2% of cancer patient group members (n = 96) were aware of gifts of stationery, 62.1% considered that gifts from the pharmaceutical industry were unethical, and 74.7% considered that these gifts would affect physicians' prescribing practices [12].

Patients' trust in physicians is essential for treatment; higher trust correlates with better health outcomes and vice versa [13]. Therefore, it is important to examine how patients' perceptions of physician-PR relationships impact their trust in physicians, which has been investigated in two studies from the U.S. [7, 8]. These studies suggested that the belief among patients that physicians receive gifts from PRs is associated with less trust in their physicians [7, 8] and the healthcare system at large [8], and with lower adherence to treatment regimens prescribed by their physicians [7]. In addition, in a study from Japan, approximately 20% and 45% of cancer patient group members (n = 96) reported decreased trust in physicians who accepted stationery and meals, respectively [12].

The purposes of this study were to clarify the public's perceptions of physicians' relationships with PRs, examine the association between their perceptions and their trust in physicians using a Japanese version of a validated scale for interpersonal trust in physicians, and compare public perceptions with those of physicians regarding the physician-PR interactions. To achieve these purposes, in parallel with a web-based survey of the public, we also conducted a nationwide mail-based survey of Japanese physicians, the latter of which is reported in detail elsewhere [14].

## Methods

### Ethics

The Ethical Committee of Kurume University (health care ethics) approved the study protocol (study number 21142). At the beginning of the survey, participants were required to provide written consent by checking 'I agree.' in response to a question, indicating their willingness to participate in the study.

### Study design and participants

A cross-sectional, self-administered, anonymous, internet-panel survey was conducted. Survey participants were residents of Japan, aged 18 years or older, capable of answering the questionnaire in Japanese, and were not healthcare professionals. The parallel study of Japanese physicians utilized a cross-sectional mail-based survey design and included Japanese practicing physicians working in clinical and hospital settings across all 47 prefectures in Japan. The participants were selected from seven specialties and included both clinic-based and hospital-based physicians [14].

### Survey implementation

The survey implementation was contracted to Cross Marketing Inc., which has an active panel of approximately 5.0 million people [15]. A cohort of 28,132 pre-registered participants in the active panel were sent an invitation e-mail with a link to the online survey webpage. Those who responded to the four screening items (gender, age, non-healthcare professional, and consent to the survey) and answered yes to the latter two screening items were allowed to proceed to the remaining 26 items. The answers were registered when they completed all 30 questions. The survey was conducted on Thursday, October 28, 2021, and was closed when the number of participants reached 1000. The survey participants were offered non-cash points as a small token of appreciation. From January to March 2021, the physician survey was conducted in parallel using the methods which involved sending participants a pre-notification postcard, followed by a package containing a questionnaire, a self-addressed postcard for response tracking, a stamped reply envelope, and a small incentive. Up to two reminders were sent to non-respondents at two-week intervals [14].

### Survey instrument

The items of the survey instrument were developed on the basis of a literature search and discussion by the first and second authors. The survey instrument was modified in terms of question wording and response format after a preliminary survey among non-medical and medical professionals in a hospital where the first author worked. The participants voluntarily answered questions and provided qualitative feedback on the questionnaire. We modified the wording of the questionnaire to allow participants to respond without guidance, ensuring the questions measured what they were intended to.

At the beginning of the survey, for inclusion of the participants, they needed to answer "agree" to a question asking whether they consented to participate in the study, as noted above. Participants were asked background information such as gender (male, female, or other), age, whether they had a primary care physician, and health status. We asked about their awareness, acceptance, and perceptions of the influence of physician-PR interactions. First, the participants were asked to answer "yes" or "no" as to whether they had ever noticed physicians using office stationery (e.g., pens) with the name of a pharmaceutical company and/or specific medication, placement of promotional materials (e.g., calendars) in

examination rooms, or visits by PRs to medical facilities during their medical consultations ("*Awareness*"). Second, they were asked about their acceptance of four types of physician-PR interactions (gifts of stationery and medical textbooks, meals provided by PRs, and participation in promotional seminars) using a 5-point Likert scale ("*Acceptability*"). Third, they were asked whether they believed that three types of physician-PR relationships (information, stationery, and meals provided by PRs) would affect physicians' prescribing behaviors using a 5-point Likert scale ("*Influence*"). Fourth, they were asked about their estimates of the proportion of physicians in general receiving stationery or meals from PRs ("*Estimates*"). They were also asked whether they would like to know the relationships between their personal physicians and the pharmaceutical industry. To measure their trust in physicians in general, we used an 11-item, 5-point scale called "Interpersonal Trust in a Physician" (score range of 11–55) originally developed by Hall et al. [16] and translated into Japanese by Katsuyama et al. [17]. We obtained permission from Katsuyama to use the scale in this study.

The mailed national physician survey, administered concurrently with the public survey, comprised a 28-item questionnaire. This questionnaire assessed background information of the respondents, pharmaceutical promotions at the workplace, frequency of involvement in promotional activities, and attitudes toward relationships with PRs. It measured involvement in promotional activities and attitudes toward relationships with PRs using Likert scales [14].

## Statistical analysis

Previous studies [7, 8] indicate that approximately half of the population believes physicians maintain relationships with pharmaceutical companies. A trust difference of 3–4 points is anticipated between individuals who hold this belief and those who don't. With the actual scale having a standard deviation of 6.07 (range 11–55), this translates to a standardized effect size of 0.49–0.66. This effect size, which quantifies the magnitude of trust difference, is derived by dividing the difference (3–4 points) by the standard deviation (6.07).To ensure our study was robustly powered, we chose a standardized effect size of 0.8 for our sample size calculation, even though previous studies indicated effect sizes between 0.49 and 0.66. Oping for 0.8, which is "large" by Cohen's convention, is expected to provide an estimate of the difference that must be detected. With this effect size, alongside an α value (two-sided) of 0.05 and a β value of 0.20, we estimated a need for 45–64 samples per group. Factoring in an expected response rate of about 10%, the calculated total sample size ranged from 900 to 1280. Consequently, we set the sample size for this survey at 1000.

The results obtained from the public survey were compared to those from the physician survey, specifically focusing on 10 questions related to awareness, acceptance, and perceptions of influence. The physician survey was anonymously conducted by mail in 2021, with 1636 valid responses (63.2% valid response rate) [14]. The items of interest for comparison in the physician survey included the physician's workplace situation (vs. *awareness*), appropriateness of involvement (vs. *acceptability*), and influence (vs. *influence*). We used $\chi^2$ test to make the statistical comparisons between the public's and physicians' awareness, acceptance, and perceptions of the influence of physician-PR interactions.

Multiple regression analysis (forced entry method) was used to determine the factors associated with the level of trust in physicians. The objective variable was the "Interpersonal Trust in a Physician" scale (score range of 11–55, higher score means higher trust), and the explanatory variables were gender, age, presence of a primary care physician, health status ("very good" = 1 to "very poor" = 5), *awareness* ("yes" = 1 or "no" = 0, sum of scores of three items; range 0–3), *acceptance* ("acceptable" = 1 to "unacceptable" = 5, sum of scores of four items; range 4–20), *influence* ("not influential" = 1 to "Very influential" = 5, sum of scores of three

items; range 3–15), and *estimates* of the proportions of physicians accepting gifts ("almost none" = 1 to "most" = 5, sum of scores of two items; range 2–10).

For all analyses, *p*-values less than 0.05 obtained by a two-tailed test were defined as statistically significant. SAS9.4 (SAS Institute Inc., Cary, NC) was used for all analyses.

## Results

### Characteristics of participants

The characteristics of 1000 participants are shown in Table 1. The mean age was 44.8 years (range 18–95, standard deviation 18.3). Forty-eight percent of the participants were female. Overall, 41.5% of the participants had a primary care physician and 83.3% of them were in fair or better health.

### Public perceptions of physician-pharmaceutical industry relationships

**Awareness.**   In examination rooms and/or waiting areas of medical facilities, 28.9% of the participants had noticed physicians using office stationery with the name of a pharmaceutical company and/or specific medication, 44.2% had noticed the placement of promotional materials (e.g., calendars), and 29.2% had noticed the presence of PRs (Table 2).

**Acceptance.**   Among the participants, 69.4%, 50.5%, 65.5%, and 23.1% believed that physicians' receiving stationery and medical textbooks, attending promotional seminars at the workplace, and receiving meals at restaurants were at least somewhat acceptable, respectively (Table 3).

**Influence.**   While only 14.8% of the participants thought that physicians' prescribing habits would be influenced by their receiving stationery, the corresponding proportions were 53.5% and 56.2% receiving information from PRs and receiving meals at restaurants, respectively (Table 4).

**Table 1. Characteristics of participants (N = 1000).**

| Characteristic | No. | % |
|---|---|---|
| Age | | |
| 18–39 years | 440 | 44.0 |
| 40–64 years | 389 | 38.9 |
| 65 years or older | 171 | 17.1 |
| Gender | | |
| Male | 517 | 51.7 |
| Female | 480 | 48.0 |
| Other | 3 | 0.3 |
| Having a primary care physician | | |
| Yes | 415 | 41.5 |
| No | 585 | 58.5 |
| Self-rated health status | | |
| Very good | 94 | 9.4 |
| Good | 275 | 27.5 |
| Fair | 464 | 46.4 |
| Poor | 139 | 13.9 |
| Very poor | 28 | 2.8 |

**Table 2. The public's awareness and physicians' reporting of physician-pharmaceutical industry relationships in examination rooms and/or waiting areas of medical facilities.**

| Type of physician-pharmaceutical industry relationships | The public aware of relationships (N = 1000) | Physicians' reporting of relationships (N = 1636) | P-value |
|---|---|---|---|
| | No. (%) | No. (%) | |
| Office stationery used | 289 (28.9) | 1008 (61.6) | < .001 |
| Promotional materials placed | 442 (44.2) | 830 (50.7) | < .001 |
| Presence of PRs | 292 (29.2) | 1450 (88.6) | < .001 |

Abbreviations: PR, pharmaceutical representative

**Estimates.** When asked about their estimates of the proportions of physicians receiving gifts from PRs, 82.2% and 56.8% of the participants thought that half or more of the physicians in general received office stationery and meals at restaurants, respectively.

**Desire to know their physicians' relationships with the pharmaceutical industry.** Thirty-two percent of the participants at least somewhat desired to know what kind of relationship their physician has with the pharmaceutical industry, while 27.4% did not.

## Factors associated with higher trust in physicians

The mean score (standard deviation) of the "Interpersonal Trust in a Physician" scale (score range of 11–55, higher score means higher trust) was 31.9 (6.53). Factors associated with higher trust in physicians included participants being 65 years or older [partial regression coefficient 1.74 (95% confidence interval 0.58 to 2.91)], having a primary care physician [3.22 (2.30 to 4.14)], being in better health [−1.42 (−1.88 to −0.95)], the belief that physicians' involvement in promotional activities is acceptable [−0.52 (−0.65 to −0.39)], and their estimate that physicians are not receiving gifts from PRs [−0.70 (−0.92 to −0.47)] (Table 5).

## Comparison with the physician survey

The proportions of our participants from the general public who were aware of physicians using office stationery with the name of a pharmaceutical company and/or specific medication, placement of promotional materials (e.g., calendars), and presence of PRs in examination rooms and/or waiting areas of medical facilities were significantly lower than those of Japanese physicians who reported that these practices actually occur (Table 2). For the acceptability of physicians' receiving stationery or medical textbooks, attending promotional drug seminars at the workplace, and receiving meals at restaurants, the public were significantly more accepting of these practices than physicians (Table 3). Meanwhile, the proportion of the public who thought that physician-PR relationships would influence their prescribing habits was higher than that of physicians (Table 4).

**Table 3. The public's attitudes regarding acceptability and physicians' attitudes regarding appropriateness of physician-pharmaceutical industry relationships.**

| Type of physician-pharmaceutical industry relationships | The public who consider it at least somewhat acceptable (N = 1000) | Physicians who consider it at least somewhat appropriate (N = 1636) | P-value |
|---|---|---|---|
| | No. (%) | No. (%) | |
| Receiving stationery with the name of a pharmaceutical company and/or specific medication | 694 (69.4) | 598 (36.6) | < .001 |
| Receiving medical textbooks | 505 (50.5) | 331 (20.2) | < .001 |
| Attending promotional drug seminars at the workplace | 655 (65.5) | 608 (37.2) | < .001 |
| Receiving meals at restaurants | 231 (23.1) | 215 (13.1) | < .001 |

**Table 4. The public's and physicians' perceptions of influence of physician-pharmaceutical industry relationships.**

|  | The public who consider it influential (N = 1000) | Physicians who consider it influential (N = 1636) | P-value |
|---|---|---|---|
|  | No. (%) | No. (%) |  |
| Receiving information from PRs | 535 (53.5) | 230 (14.1) | < .001 |
| Receiving stationery with the name of a pharmaceutical company and/or specific medication | 148 (14.8) | 35 (2.1) | < .001 |
| Receiving meals at restaurants | 562 (56.2) | 171 (10.5) | < .001 |

Abbreviations: PR, pharmaceutical representative

## Discussion

To the best of our knowledge, this is the first large-scale survey conducted in Japan that sought to clarify the public perceptions of physician-PR relationships and their association with trust in physicians. Many of our participants were less aware of certain physician-PR relationships. The public was more accepting of physicians' receiving stationery and/or medical textbooks and attending promotional drug seminars at their workplaces compared to receiving meals at restaurants. Many thought that the physicians' involvement in promotional activities led by PRs influenced their prescribing habits and estimated that the majority of physicians received office stationery and meals from PRs. They were divided regarding the desire to know about their physicians' relationships with the industry. Factors associated with higher trust in physicians included age (65 years or older), having a primary care physician, better health status, the belief that physicians' involvement in promotional activities is acceptable, and their estimation that most physicians are not receiving gifts from PRs. Compared to the physicians, the public had lower awareness of and higher acceptance of physicians' involvement in promotional activities. On the other hand, the public believed that physician-PR relationships influenced physicians' prescribing habits more than the physicians themselves.

In our study, fewer participants were aware of office stationery in examination rooms and waiting areas compared to prior U.S. studies [8, 18, 19]. However, their awareness of promotional items such as calendars and the presence of PRs was consistent with a previous study [8]. Regarding the acceptability of gifts to physicians, our participants' views on receiving stationery, medical textbooks, and meals at restaurants were in line with earlier findings [9, 11, 18, 20]. They believed that stationery gifts had less influence on physicians than perceived by U.S. patients [11], but more than the views of patients in Turkey [21]. Additionally, our

**Table 5. Multivariate analysis of higher trust in physicians.**

| Explanatory variables |  | Coefficient | 95% confidence interval | P-value |
|---|---|---|---|---|
| Gender | Male (= 1) vs. female (= 0) | 0.51 | −0.32 to 1.35 | 0.23 |
| Age | ≥65 years (= 1) vs. <65 years (= 0) | 1.74 | 0.57 to 2.91 | 0.0035 |
| Having a primary care physician | Yes (= 1) vs. no (= 0) | 3.22 | 2.30 to 4.14 | < .0001 |
| Self-rated health status | Very good (= 1) to very poor (= 5) | −1.42 | −1.88 to −0.95 | < .0001 |
| Perceptions |  |  |  |  |
| *Awareness* | Never aware = 0; range 0–3 | −0.11 | −0.51 to 0.29 | 0.59 |
| *Acceptance* | Acceptable = 0; range 0–4 | −0.52 | −0.65 to −0.39 | < .0001 |
| *Influence* | Least perceived influence = 0; range 0–3 | 0.14 | −0.07 to 0.34 | 0.19 |
| *Estimates* | No involvement = 0; range 0–2 | −0.70 | −0.92 to −0.47 | <0.0001 |

The objective variable was the score on the Interpersonal Trust in a Physician scale (range 11–55). Higher score means higher trust.

participants felt that gifts of meals influenced physicians more than the beliefs held by both U. S. and Turkish patients [11, 21]

Despite the observed variations in public or patient perceptions of physicians' involvement in PR-led promotional activities across these studies, many participants were critical or skeptical of such activities.

Overall, compared to physicians in our separate surveys, the public showed less awareness of physician-PR relationships, yet appeared more accepting of these relationships and more critical of their potential influence. Future research is necessary to understand why the Japanese public appears relatively accepting of these relationships, while simultaneously expressing more critical views about their potential consequences.

This study has several limitations. First, due to the inherent sampling bias in internet-panel research [15], our participants might not accurately represent the broader Japanese public. Out of 28,132 pre-registered individuals in the active panel, the first 1,000 to respond were selected for our study. Notably, our participant group was younger compared to the overall age distribution of Japanese adults, which could limit the generalizability of our findings, particularly regarding the opinions of older adults. Second, there is a possibility of social desirability bias. Although our survey was conducted online without collecting personally identifiable information—reducing the likelihood of such bias compared to face-to-face or mail-based surveys [22]–the potential bias cannot be completely ruled out. Third, due to the cross-sectional design of our survey, we cannot infer causality from the associations observed. Fourth, our results may not be universally applicable to populations outside Japan due to cultural and healthcare-related differences. For example, the Japanese healthcare system exhibits unique characteristics, including universal health insurance coverage for all citizens, a lack of a robust primary care system, a small number of certified generalist physicians, and unrestricted access to specialist physicians without a referral. Additionally, there tends to be less public discourse regarding the relationships between physicians and the pharmaceutical industry in Japan, including in the lay media, compared to the U.S. and other countries.

Our survey provided insights into Japanese public perceptions of physician-pharmaceutical industry relationships and their impact on trust in physicians. Physicians should be aware of these perceptions and carefully consider how to foster appropriate relationships with the industry. Failing to do so could risk eroding patient trust and public confidence.

## Supporting information

**S1 File. The survey instrument in English.**
(DOCX)

## Acknowledgments

We would like to thank Cross Marketing Inc. for implementing the survey. We would like to thank Edanz (https://jp.edanz.com/ac) for editing a draft of this manuscript.

## Author Contributions

**Conceptualization:** Sayaka Saito, Kei Mukohara.

**Data curation:** Sayaka Saito.

**Formal analysis:** Kazuhiro Shimomura, Kenta Murotani.

**Funding acquisition:** Kei Mukohara.

**Methodology:** Sayaka Saito, Kei Mukohara.

**Supervision:** Kei Mukohara.

**Writing – original draft:** Sayaka Saito.

**Writing – review & editing:** Kei Mukohara, Kazuhiro Shimomura, Kenta Murotani.

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
