## [Decision Letter · Decision Letter 0]

1 Sep 2023

PONE-D-23-18061Public perceptions of physicians-pharmaceutical industry relationships and trust in physiciansPLOS ONE

Dear Dr. Mukohara,

Thank you for submitting your manuscript to PLOS ONE. After careful consideration, we feel that it has merit but does not fully meet PLOS ONE’s publication criteria as it currently stands. Therefore, we invite you to submit a revised version of the manuscript that addresses the points raised during the review process.

We look forward to receiving your revised manuscript.

Kind regards,

Soham Bandyopadhyay

Academic Editor

PLOS ONE

Journal Requirements:

4. Please expand the acronym “JSPS” (as indicated in your financial disclosure) so that it states the name of your funders in full. This information should be included in your cover letter; we will change the online submission form on your behalf.

Reviewers' comments:

Reviewer's Responses to Questions

**Comments to the Author**

1. Is the manuscript technically sound, and do the data support the conclusions?

Reviewer #1: Yes

2. Has the statistical analysis been performed appropriately and rigorously? 

Reviewer #1: Yes

3. Have the authors made all data underlying the findings in their manuscript fully available?

Reviewer #1: Yes

4. Is the manuscript presented in an intelligible fashion and written in standard English?

Reviewer #1: Yes

5. Review Comments to the Author

Reviewer #1: Dear colleague(s)

Thank you for this interesting and well-written paper on public perceptions of physicians-pharmaceutical industry relationships and trust in physicians.

It covers an important topic and represents a significant amount of work done by the authors.

The introduction summarises the area of work of the study. The objective of the paper/study is also clearly described and is relevant in addressing the gap identified.

The rationale for the paper is outlined and the results are interesting.

In the methods section, how the sample was recruited, the use of the items of interest to gauge perceptions on physicians-PR relationship and trust in physicians regarding appropriateness in gift-taking were well-described.

However, some sections in the results and discussion should be better presented and explained. Comparisons of the results with previous studies from the U.S and Turkey for example, could be made clearer and less complicated. In addition, the implications of the study towards physicians' prescription habit could be further fleshed out, as the the paper only hints out on the role played by the Japanese culture and its healthcare system.

In short, this paper (albeit minor corrections) is a valuable addition to the knowledge repository pertaining to this subject.

6. PLOS authors have the option to publish the peer review history of their article (what does this mean?). If published, this will include your full peer review and any attached files.

Reviewer #1: No

---

## [Author Response · Author response to Decision Letter 0]

5 Nov 2023

Dear Dr. Soham Bandyopadhyay,

We are submitting a revised version of our manuscript titled “Public perceptions of physicians-pharmaceutical industry relationships and trust in physicians” (reference number PONE-D-23-18061) for consideration for publication in PLOS ONE.

We are grateful for the constructive comments provided by the reviewer. These suggestions have been instrumental in enhancing the quality of our manuscript. We have addressed each comment in detail. All revisions have been highlighted in yellow, and any deletions are indicated with a strikethrough. We have also inserted our comments in the manuscript addressing both the corrections and comments made by the reviewer.

We hope that our revised manuscript meets the publication standards of PLOS ONE.

Sincerely,

Kei Mukohara, MD

 

Reviewer #1: Comparisons of the results with previous studies from the U.S and Turkey for example, could be made clearer and less complicated.

Response: Thank you for your feedback regarding the clarity of comparisons between our results and previous studies from the U.S. and Turkey. Based on your feedback, we revised the sentences as follows: " In our study, fewer participants were aware of office stationery in examination rooms and waiting areas compared to prior U.S. studies [8,18,19]. However, their awareness of promotional items such as calendars and the presence of PRs was consistent with a previous study [8]. Regarding the acceptability of gifts to physicians, our participants' views on receiving stationery, medical textbooks, and meals at restaurants were in line with earlier findings [9,11,18,20]. They believed that stationery gifts had less influence on physicians than perceived by U.S. patients [11], but more than the views of patients in Turkey [21]. Additionally, our participants felt that gifts of meals influenced physicians more than the beliefs held by both U.S. and Turkish patients [11,21] " 

Reviewer #1: In addition, the implications of the study towards physicians' prescription habit could be further fleshed out, as the paper only hints out on the role played by the Japanese culture and its healthcare system.

Response: We appreciate your insights. Our study primarily aimed to compare public perceptions with those of physicians on physician-industry relationships. We mentioned the Japanese culture and healthcare system in the limitations section to highlight potential constraints in the generalizability of our findings and not to suggest direct influences on physicians' prescribing habits.

---

## [Editor Report · Decision Letter 1]

9 Nov 2023

Public perceptions of physician-pharmaceutical industry relationships and trust in physicians

PONE-D-23-18061R1

Dear Dr. Mukohara,

We’re pleased to inform you that your manuscript has been judged scientifically suitable for publication and will be formally accepted for publication once it meets all outstanding technical requirements.

Kind regards,

Soham Bandyopadhyay

Academic Editor

PLOS ONE

---

## [Editor Report · Acceptance letter]

16 Nov 2023

PONE-D-23-18061R1 

Public perceptions of physician-pharmaceutical industry relationships and trust in physicians 

Dear Dr. Mukohara:

I'm pleased to inform you that your manuscript has been deemed suitable for publication in PLOS ONE. Congratulations! Your manuscript is now with our production department. 

Kind regards, 

on behalf of

Dr. Soham Bandyopadhyay 

Academic Editor

PLOS ONE